

# Extreme heat in India and anthropogenic climate change

Geert Jan van Oldenborgh[1], Sjoukje Philip[1], Sarah Kew[1], Michiel van Weele[1], Peter Uhe[2,7], Friederike Otto[2], Roop Singh[3], Indrani Pai[4,5], and Krishna AchutaRao[6]

[1]Royal Netherlands Meteorological Institute (KNMI), De Bilt, Netherlands
[2]Environmental Change Institute, University of Oxford, Oxford, U.K.
[3]Red Cross Red Crescent Climate Centre, The Hague, Netherlands
[4]Columbia Water Center, Columbia University, New York, New York, USA
[5]Climate Central, Princeton, U.S.
[6]Indian Institute of Technology Delhi, New Delhi, India
[7]Oxford e-Research Centre, University of Oxford, Oxford, U.K.

*Correspondence to:* G. J. van Oldenborgh (oldenborgh@knmi.nl)

**Abstract.** On 19 May 2016 the afternoon temperature reached 51.0 °C in Phalodi in the northwest of India, a new record for the highest observed maximum temperature in India. The previous year, a widely-reported very lethal heat wave occurred in the southeast, in Andhra Pradesh and Telangana, killing thousands of people. In both cases it was widely assumed that the probability and severity of heat waves in India are increasing due to global warming, as they do in other parts of the world. However, we do not find positive trends in the highest maximum temperature of the year in most of India since the 1970s (except spurious trends due to missing data). Decadal variability cannot explain this, but both increased air pollution with aerosols blocking sunlight and increased irrigation leading to evaporative cooling have counteracted the effect of greenhouse gases up to now. Current climate models do not represent these processes well and hence cannot be used to attribute heat waves in this area.

The health effects of heat are often described better by a combination of temperature and humidity, such as a heat index or wet bulb temperature. Due to the increase in humidity from irrigation and higher SSTs these indices have increased over the last decades even when extreme temperatures have not. The extreme air pollution also exacerbates the health impacts of heat. From a health impact point of view, the severity of heat waves has increased in India.

For the next decades we expect the trend due to global warming to continue, but the cooling effect of aerosols to diminish as air quality controls are implemented. The expansion of irrigation will likely continue, though at a slower pace, mitigating this trend somewhat. Humidity will probably continue to rise. The combination will give a strong rise of the temperature of heat waves. The high humidity will make health effects worse, whereas decreased air pollution would decrease the impacts.

## 1 Introduction

In India, the highest temperatures occur before the monsoon starts, typically in May or the beginning of June. Especially daily maximum temperatures are very high during that time. In the arid areas in the northwest, afternoon temperatures often rise into the high 40s. On 19 May 2016 temperatures exceeded 50 °C in a region on the India-Pakistan border. In Phalodi the



temperature even reached 51.0 °C, which is India's all time record (see also Fig. 1a-c). The previous record, from 1956, was 50.6 °C. The heat wave lasted three days, with temperatures the days before and after the hottest day within one degree of that value.

Excessive heat can have a devastating impact on human health, resulting in heat cramps, exhaustion, and life-threatening
heat strokes. Children and the elderly are most vulnerable. It can also aggravate pre-existing pulmonary conditions, cardiac conditions, kidney disorders and psychiatric illness. High air pollution in India exacerbates many of these problems. According to newspaper reports, at least 17 heat-related deaths occurred in the Gujarat state, 7 in Madhya Pradesh and 16 in the state of Rajasthan, where the highest temperatures were recorded during the heat wave around 19 May 2016 (e.g., http://timesofindia. indiatimes.com/city/bhopal/Heat-stroke-kills-7-in-Madhya-Pradesh/articleshow/52403498.cms). Hundreds more people were
admitted to hospitals in western India with signs of heat-related illness.

The May 2016 temperature record followed a severe heat wave in Andhra Pradesh and Telangana in May 2015, which although not record-high in temperature had a large humanitarian impact with at least 2,422 deaths attributed to the heat wave by local authorities (http://ndma.gov.in/images/guidelines/guidelines-heat-wave.pdf), more than half of which occurred in Andhra Pradesh. It is likely that the actual number of deaths is much higher as it is difficult to attain figures from rural areas
and deaths due to conditions that are exacerbated by the heat (e.g., kidney failure, heart disease) are often not counted (Azhar et al., 2014). Those directly exposed to the heat including outdoor workers, the homeless and those with pre-existing medical conditions (e.g., the elderly) constitute the majority of negative heat-related outcomes in India (Tran et al., 2013; Nag et al., 2009).

Naturally, the question was raised whether human-induced climate change played a role in this record breaking heat. While
the trend in global average temperatures in general increases the probability of heat waves occurring (Field et al., 2012; Stocker et al., 2013), this does not mean that heat waves in all locations are becoming more frequent, as factors other than greenhouse gases also affect heat. In this article we investigate the influence of anthropogenic factors on the 2016 heat wave in Rajasthan, northwestern India, and the 2015 heat wave in Andhra Pradesh and Telangana, eastern India, which are shown in Fig. 1.

There are many different definitions of heat waves. Most meteorological organisations have very different official definitions,
tailored to local conditions and stakeholders, usually based on maximum temperature and duration. In more sophisticated definitions, the temperature and duration index may be accompanied by humidity, as humid heat waves pose a greater threat to human health (Gershunov et al., 2011). A simple measure that includes this is the wet bulb temperature, which is the lowest temperature a body can attain by evaporation. It is therefore a measure of how well the human body can cool itself via evaporation of sweat from the skin. As an example, a temperature of 50 °C with a relative humidity of 40% has a wet
bulb temperature of 36 °C. This means it is equivalent to 36 °C at 100% humidity, which is a condition in which it is almost impossible for the human body to cool itself.

The Indian Meteorological Department (IMD) uses complicated definitions of heat waves and severe heat waves based on single-day maximum temperature (imd.gov.in/section/nhac/termglossary.pdf):

1. Heat Wave need not be considered till maximum temperature of a station reaches at least 40 °C for Plains and at least 30
°C for Hilly regions





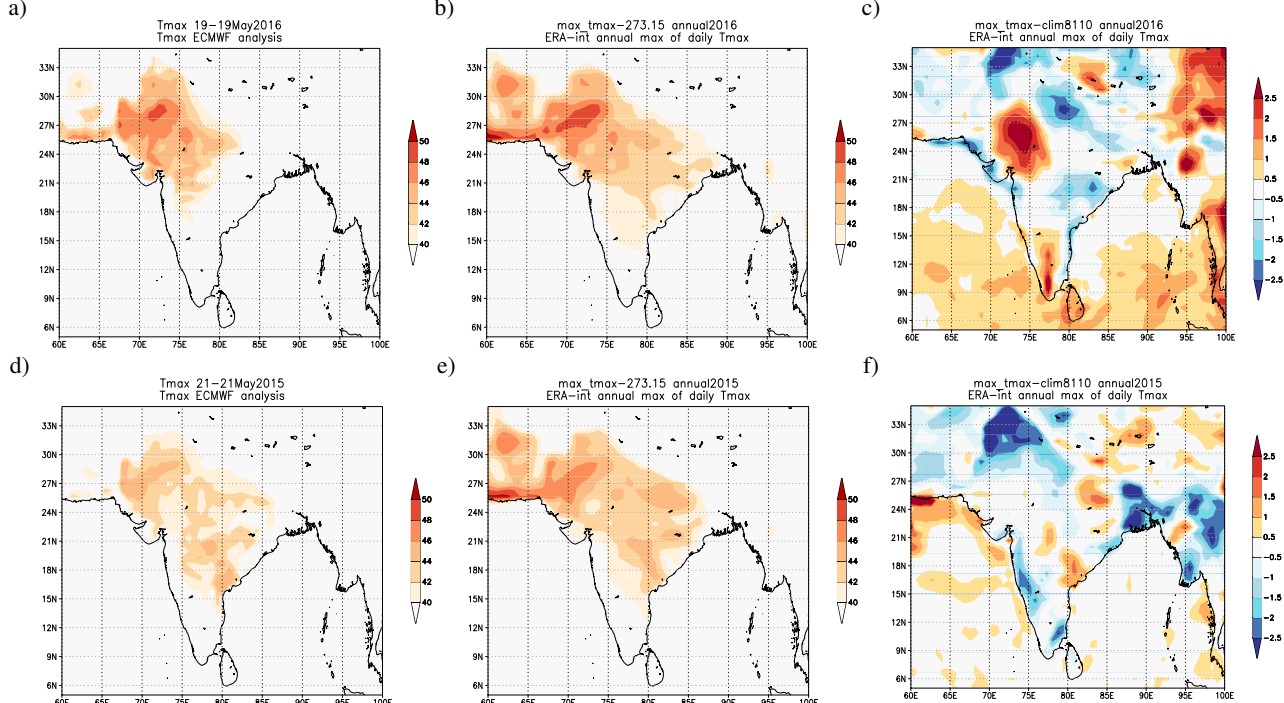

**Figure 1.** a) ECMWF operational analysis of the daily maximum temperature on 19 May 2016 (°C), b) ERA-interim highest maximum temperature of the year, TXx, in 2016 (°C). c), Anomalies of TXx in 2016 (K). d) Same for 21 May 2015, showing the heat wave in Andhra Pradesh and Telangana, e) TXx in 2015, f) anomalies of TXx in 2015.

2. When normal maximum temperature of a station is less than or equal to 40 °C Heat Wave Departure from normal is 5 °C to 6 °C Severe Heat Wave Departure from normal is 7 °C or more

3. When normal maximum temperature of a station is more than 40 °C Heat Wave Departure from normal is 4 °C to 5 °C Severe Heat Wave Departure from normal is 6 °C or more

5    4. When actual maximum temperature remains 45 °C or more irrespective of normal maximum temperature, heat waves should be declared.

Four recent studies investigated trends in heat waves in India (Pai et al., 2013; Jaswal et al., 2015; Rohini et al., 2016; Wehner et al., 2016). Using the IMD definitions, Pai et al. (2013) studied the trend in (severe) heat waves over India using station data from 1961–2010. In north, northwest and central India, some stations showed a significant increase in trend in (severe) heat

10   wave days. However, other stations showed a significant decreasing trend in (severe) heat waves. The station of Phalodi in Rajasthan state, the site of the 2016 record, showed a non-significant positive linear trend in maximum temperature anomaly over the hot weather season over 1961–2010. Overall, no consistent long term trends were observed in heat wave days over the whole country.




Another heat wave criterion considers the serious effects on human health and public concerns when the daily maximum temperature exceeds the human core body temperature (i.e., neglecting the cooling effects of perspiration). This criterion is used by Jaswal et al. (2015), who use a threshold of 37 °C during the summer season Mar-Jun. Using data from 1969–2013, their findings indicate that long period trends show an increase in summer high temperature days in north, west, and south regions, and a decrease in north-central and east regions. This does not, however, give information about the height of the maximum temperatures. In Rajasthan, a maximum temperature of 37 °C is a cool day in May.

More recently, Rohini et al. (2016) discusses the Excessive Heat Factor, which is based on two excessive heat indices. The first is Excess Heat: unusually high heat arising from a daytime temperature that is not sufficiently discharged overnight due to unusually high overnight temperatures. The second index considered is Heat Stress: a short term (acclimatization) temperature anomaly. Using a gridded dataset from 1961–2013 they find, over a limited region in central and northwestern India, that frequency, total duration and maximum duration of heat waves are increasing. However, in the rest of India they find no significant trend.

As we were writing this article, Wehner et al. (2016) was published investigating the anthropogenic influence on the May 2015 Andhra Pradesh / Telangana and June 2015 Karachi heat waves in 1- and 5day mean daily maximum of temperature and heat index. The latter also includes humidity. They find little trend in the temperatures in Karachi, but a strong trend in Hyderabad. The heat index has strong positive trends at both stations. This is confirmed using CESM runs in the current climate and a counterfactual climate without anthropogenic emissions. Again the difference is much more pronounced in the heat index than in temperature.

In this article we mainly use a very simple definition: the highest daily maximum temperature of the year, TXx (Karl et al., 1999). This is related to the IMD definition of heat waves but rather than a simple dichotomy it is a continuous measure that also describes the severity of the heat wave. It is thus also amenable to extreme value statistics. We mainly focus on the area of the 2016 record heat wave but also mention other regions, notably the location of the 2015 heat wave in Andhra Pradesh and Telangana. As society is adapted to the weather of that location, we also show the anomalies relative to a long-term (1981–2010) mean of TXx. The second definition we use is the monthly maximum of the daily maximum of the wet bulb temperature (Sullivan and Sanders, 1974) as a measure that combines heat and humidity and indicates how well the body can dissipate heat through perspiration. This is related to, but not the same as, the heat index of (Wehner et al., 2016).

We start with observed temperatures in 2015 and 2016, and trends in observed temperatures, with a detour to the effect of missing data on these trends. Next we discuss three factors that may have influenced heat waves besides global warming due to greenhouse gases: decadal variability, aerosol trends and changes in humidity. The combination is investigated further in global coupled climate models from the Coupled Model Intercomparison Project Phase 5 (CMIP5) and a large ensemble of SST-forced models. At the end we synthesise our findings into a an qualitative overview of anthropogenic forcings on the heat waves.




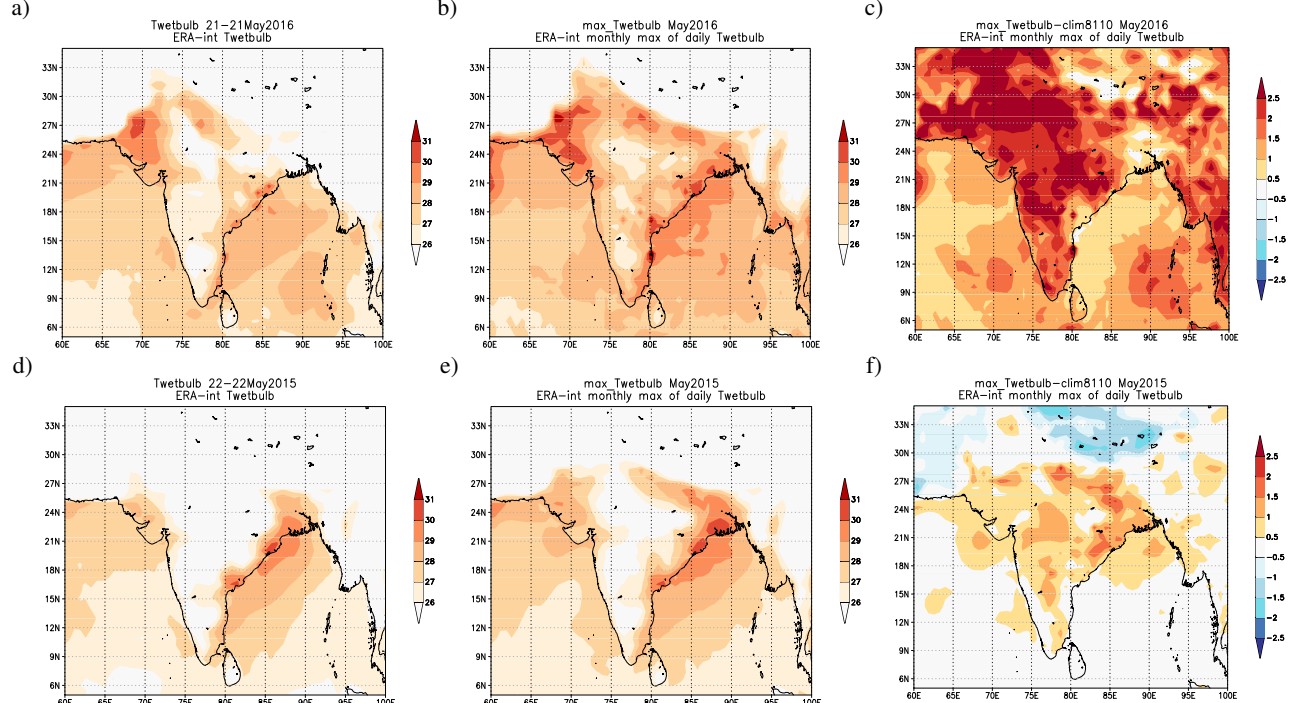

**Figure 2.** a) ERA-interim wet bulb temperature on 21 May 2016. b) Monthly maximum of the wet bulb temperature in May 2016 (°C). c) Anomalies of the maximum wet bulb temperature in May 2016 (K), see text for note. d,e,f) Same for 22 May 2015.

## 2   Temperature observations

The record maximum temperature in India was observed on 19 May 2016. The maximum temperature of the ECMWF analysis for that day is shown in Fig. 1a. The analysis underestimates the heat somewhat relative to the in-situ observations. For the northwestern part of India the temperature map of 19 May 2016 is comparable to the temperature map of TXx for 2016 in ERA-interim (Dee et al., 2011), Fig. 1b (which only became available in September and has lower resolution). The highest temperatures occurred in and around the northwest Indian state of Rajasthan and also in East Pakistan. The largest anomalies were recorded slightly further east, Fig. 1c, which was also mentioned in impact reports.

The Andhra Pradesh and Telangana heat wave reached temperatures of 44–45 °C on 21 May 2015 (Figs. 1d,e). This temperature is not exceptional for other regions in India, but about 1.5 °C above the normal hottest afternoon of the year there (Fig. 1f).

Next we consider the combination of heat and humidity in the wet bulb temperature. The heat was very dry on 19 May 2016 and hence wet bulb temperatures were not extremely high. However, on other days during his heat wave, such as 21 May, wet bulb temperatures reached more than 30 °C from this region to the coast (Fig. 2a,b), making heat stress a danger in outdoor labour. (Note that wet bulb temperatures there were even higher in June. The east coast also experienced similar wet bulb temperatures this year.) The wet bulb temperature was very high in 2016 compared to other years (Fig. 2c). This could be due





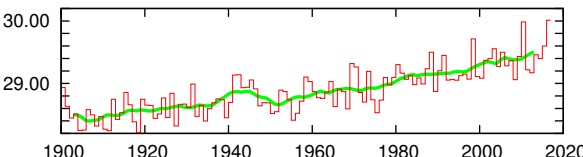

**Figure 3.** March–May mean sea surface temperature in the northern Indian Ocean (EQ–30 °N, 60–100 °E) (°C). Source: ERSST v4 (Huang et al., 2015)

to the record warm Indian Ocean, due to the trend from global warming (Bindoff et al., 2013) and the strong 2015/16 El Niño (Fig. 3). However, we can not exclude an inhomogeneity in the data from which it is computed, the ERA-interim maximum and dew point temperatures.

In 2015, the wet bulb temperatures were somewhat higher than normal in the region of the heat wave, but not by much (Fig. 2f). Both the anomalies and values of the wet bulb temperature were higher further north along the coast in Odisha and West Bangal, with values over 30 °C in the region of Kolkata (Fig. 2e). Only during the peak of the heat wave in Andhra Pradesh and Telangana were the wet bulb temperatures higher there than in West Bengal, but not higher than in Odisha.

## 3 Temperature trends

In line with global warming, the Indian annual mean temperature shows a clear trend (see, e.g., Hartmann et al., 2013). However, as pointed out by various studies (e.g., Pai et al., 2013; Rohini et al., 2016; Padma Kumari et al., 2007), this does not hold for the hot extremes. The highest maximum temperature of the year, TXx, does not show a consistent significant trend over the whole country. In some parts of India there is even a negative trend. This is shown in Fig. 4a for 1970–2012, with trends based on the gridded maximum temperature analysis from IMD (Srivastava et al., 2008). (Possible problems with gridded datasets in the study of extremes are discussed below.) The ERA-interim reanalysis, which assimilates both station data and satellite data, shows similar though lower trends over the later period 1979–2015 (Fig. 4b).

Focusing on the region of the 2016 heat wave, the public GHCN-D v3.22 dataset (Menne et al., 2012, 2016) does not contain the Phalodi series. There are two nearby stations with enough data to analyse. Bikaner (28.0 °N, 73.3 °E) has a relatively long and complete time series. It recorded a temperature of 49.5 °C on 19 May 2016 according to newspaper reports, a record relative to the GHCN-D series. However, the 2016 data are not publicly available. Jodhpur (26.3 °N, 73.2 °E), does have 2016 data with a temperature of 48.0 °C that day and 48.8 °C on 20 Jul 2016. However, the historical series is more incomplete. We analyse both series.

Fig. 5 shows the daily maximum temperature series for Jodhpur, albeit with some missing data, as well as the continuous series from ERA-interim interpolated at Jodhpur's coordinates. Together these data series indicate a heat wave duration of 3 to 4 days, with 2 days (19–20 July) reaching the 'severe' category, according to the IMD heat wave definition. Note that before 1983, temperatures are recorded in whole numbers, but in tenths of degrees Celsius after that.



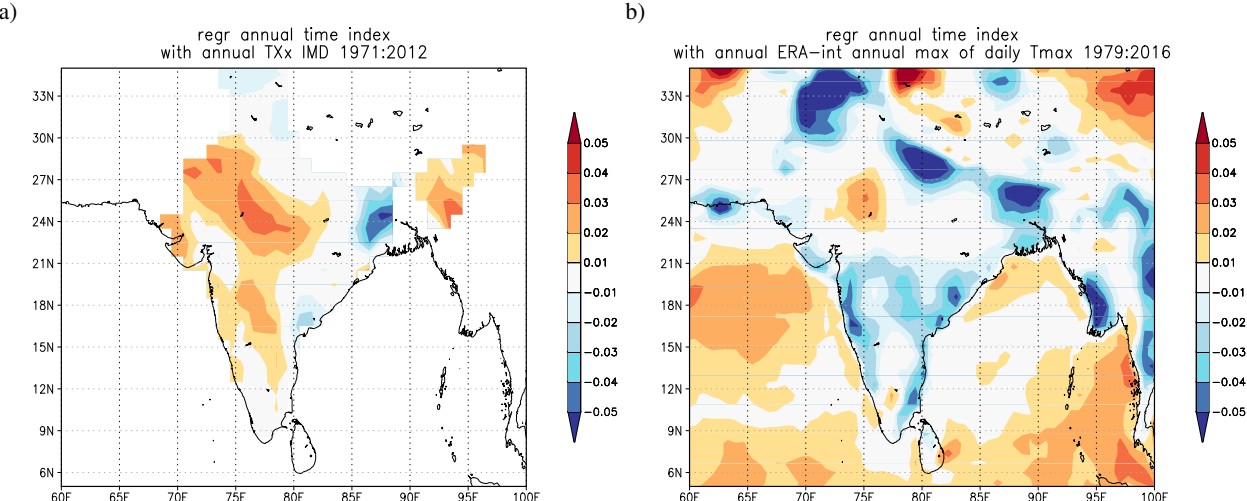

**Figure 4.** a) Trend in the highest maximum temperature in the IMD gridded daily analysis 1971–2012 (K/yr). b) Same in the ERA-interim reanalysis 1979–2015.

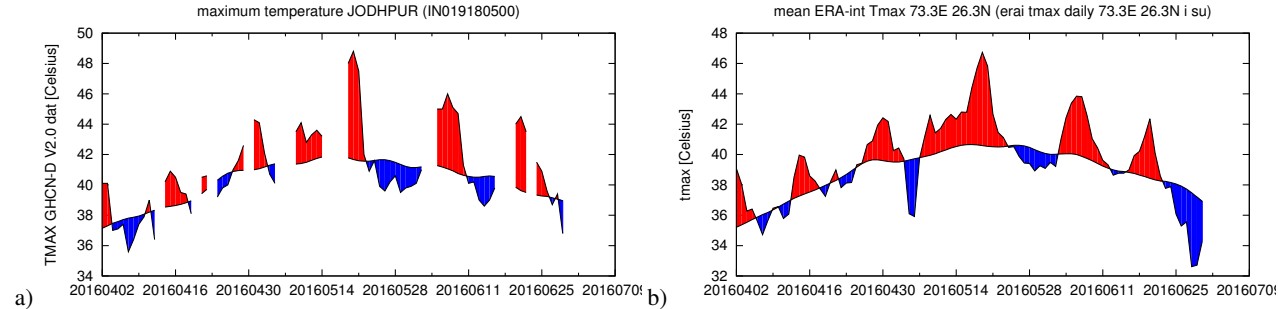

**Figure 5.** April–June daily maximum temperature time series from (a) GHCN-D v3.22 observations at Jodhpur, the closest station to Phalodi with publically available data, (b) ERA-interim interpolated at the coordinates of Jodhpur (°C). Departures from each data set's climatology (1981-2010) are shown in red (positive) and blue (negative). Days with missing data are left white.





Next we analyse the trend up to 2015. This excludes the heat wave itself, as that would give a positive bias. According to extreme value statistics theory, the May–June maxima should be distributed according to a Generalised Extreme Value function (Coles, 2001):

$$F(x) = \exp\left[ -\left( 1 + \xi \frac{x - \mu}{\sigma} \right)^{1/\xi} \right],$$ (1)

with $\mu$ the position parameter, $\sigma$ the scale parameter and $\xi$ the shape parameter. In order to incorporate possible effects of climate change we add the possibility that the position parameter changes linearly with time with a trend $\alpha$:

$$\mu = \mu_0 + \alpha t.$$ (2)

The uncertainties were estimated with a 1000-member non-parametric bootstrap procedure.

The Bikaner series starts with some data around 1958, and has more or less continuous data starting in 1973. The series contains 11.5% missing data, in 1973–2015, mainly before 2000. We demand at least 70% valid data in May–June to determine TXx, a higher threshold eliminates the observation of 49 °C in 1973. The missing data will depress TXx somewhat as it may have fallen on a day without valid observation. This effect is stronger in the earlier part of the series with more missing data. The lower TXx in the earlier data leads to a spurious positive trend. For serially uncorrelated data, the trend from 30% to almost 0% missing data would give a spurious increase in probability of a factor 1.4, simply because at the beginning of the series the probability of *observing* the extreme would be only 70%. The increase of the observed fraction from 70% to 100% looks like an apparent rise in probability of extremely high temperatures. This increase in probability corresponds to a spurious trend in temperature of roughly 0.1 K/10yr here. However, this is not the full story as the temperature values are strongly correlated from day to day: a heat wave usually lasts a few days. This implies that even if the peak was not recorded, the chance is high that one of the hot days is in the observed record. To take this day-to-day autocorrelation into account, a Monte Carlo procedure using 100 time series of random numbers with the same mean, variance, autocorrelation and missing data as the original was performed, under the assumption that the missing data is randomly distributed over the series (we verified that the missing data are not clustered). We find that the overestimation is $0.09 \pm 0.03$ K/10yr for the Bikaner series when demanding at least 70% valid data in May–June. For Jodhpur it is negligible, $0.00 \pm 0.03$ K/10yr.

Finally we are in a position to estimate the trends in TXx from the observed time series at the stations of Bikaner and Jodhpur. For this we determined TXx for each year with enough data and fitted these to Eqs 1,2 up to 2015. The results are shown in Fig. 6. The fitted trend is $0.01 \pm 0.28$ K/10yr (95% uncertainty margins) at Bikaner, $-0.15 \pm 0.30$ K/10yr at Jodhpur. Neither is significantly different from zero. They are in fact both slightly negative after subtracting the spurious trend due to the varying amount of missing data discussed above. The absence of a positive trend remains when more valid data is demanded, e.g., 80% or 90%.

The return period diagrams Fig. 6c,d show that the observed values have return periods of more than 40 yr (95% confidence interval), given the low number of data points it is impossible to say how much more. It was a rare event at that location given the past climate.




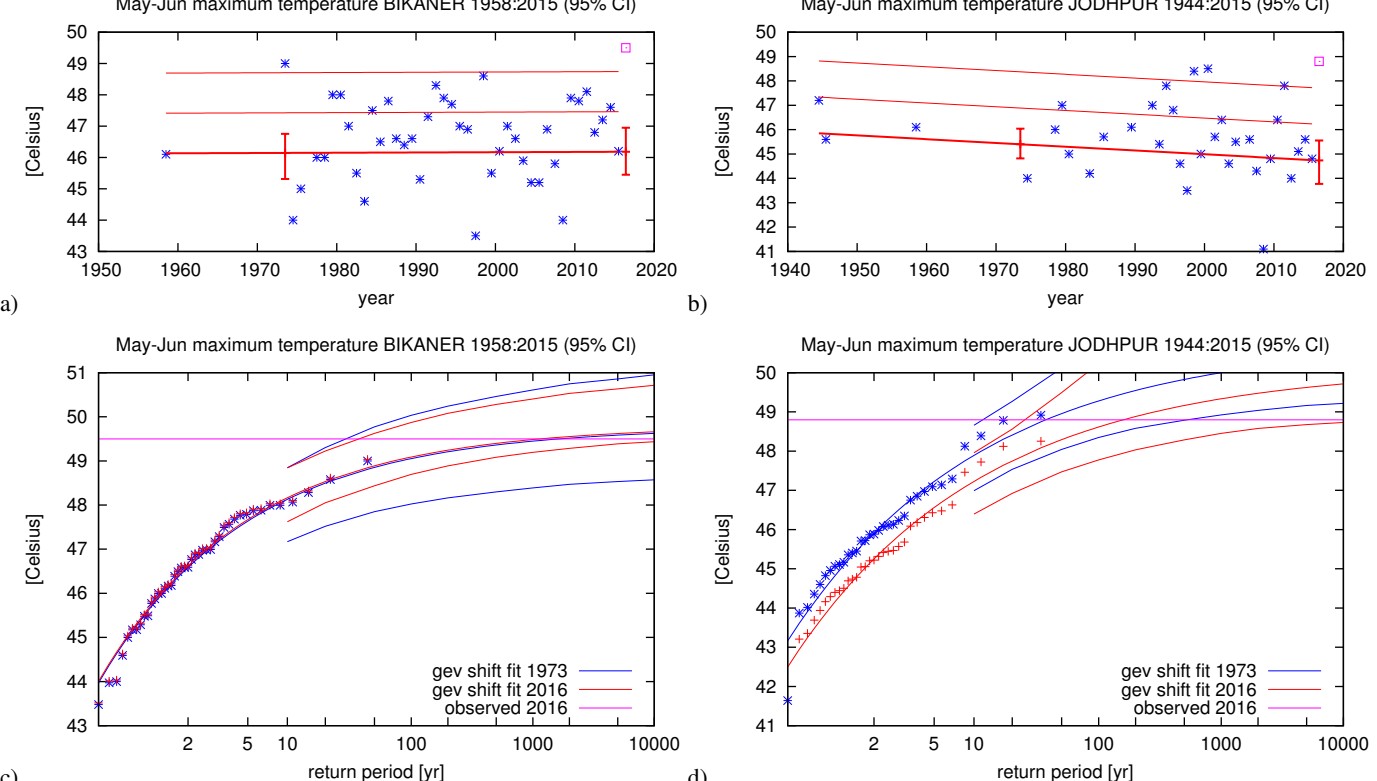

**Figure 6.** a) Observed TXx at Bikaner, Rajasthan, India (GHCN-D v3.22) with a GEV that shifts with time fitted, demanding 70% valid data in May–June. The thick line denotes $\mu$ and the thin lines $\mu + \sigma$ and $\mu + 2\sigma$. b) Same for Jodhpur. c) Gumbel plot of the fit in 1973 and in 2016 (central lines). The upper and lower lines denote the 95% confidence interval. The observations are shown twice, shifted up and down with the fitted trend. d) Same for Jodhpur.

We performed the same analysis for the heat wave in Andhra Pradesh and Telangana on 23–24 May 2015. The station of Machilipatnam is close to the centre of the heat wave and has a relatively good time series 1957–1958 and 1976–2016, with 20% missing data in the earlier part of the series, decreasing to less than 5% in recent years. The fit (not shown) for a cut-off of 70% valid data in May–June gives a non-significant trend of $0.15 \pm 0.40$ K/10yr. Repeating the procedure with 100 Monte Carlo samples with the same mean, standard deviation, autocorrelation and missing dates but no trend gives a spurious trend of $0.19 \pm 0.11$ K/10yr, so even this small trend is mostly due to the trend in missing values. Demanding 80% valid data, the observed trend becomes $0.02 \pm 0.54$ K/10yr, of which $0.13 \pm 0.07$ is due to the trend in missing values.

This agrees partially with the analysis of Wehner et al. (2016), who find no trend in Karachi, Pakistan but a positive trend in Hyderabad, Telengana, India over 1973–2014. However, the location of Hyderabad Airport in the IND gridded dataset of TXx only shows a trend in the period before 1980. Over 1973–2013 a linear trend is small and not significantly different from zero. Over 1979–2013 it is zero, whereas the ERA-interim grid point has a significant negative trend over that period.





The return period of the 2015 event at Machilipatnam is quite low, about 15 years (95% CI 9 to 40 yr), in agreement with the unexceptional temperature anomalies in Fig. 1f. In fact, given that it covered less than 1/15th of the area of India one expects a heat wave with a return period like this almost every year somewhere in the country.

The spurious trends due to trends in the fraction of missing data may also explain part of the difference in trends between the IMD observation-based TXx analysis and ERA-interim in Fig. 4. The IMD dataset is filled in by interpolating in time and/or space. An interpolated value will always be smaller than the observations it is based upon, so the more points that are interpolated rather than observed, the lower the extremes. This is not the case for the reanalysis, where the physical interpolation using a weather model of all available in situ and remote observations can also generate extremes when the ground temperature observations are missing.

We conclude that there are no significant trends for the highest temperature of the year, TXx, in the regions with the record temperatures in 2015 and 2016. Instead, we find near-zero trends. This is in contrast to most studies of heat waves in the rest of the world. For instance, for Australia, Perkins et al. (2014) show that the frequency and intensity in the likelihood of the extreme Australian heat during the 2012/13 summer had increased due to human activity. This is confirmed by Cowan et al. (2014), who find an increased likelihood in frequency and duration in the CMIP5 ensemble in Australia. Sun et al. (2014) show that there is an increase in likelihood of extreme summer heat in Eastern China. The likelihood of a given unusually high summer temperature being exceeded in North America was simulated to be about 10 times greater due to anthropogenic emissions by Rupp et al. (2015), although the observations show no trends over the eastern half since the 1930s (Peterson et al., 2013). For Central Europe, Sippel et al. (2016) use both observations and models to show that the frequency of heat waves has increased. In a Swiss study, Scherrer et al. (2016) use over 100 years of homogenized daily maximum temperature data from nine MeteoSwiss stations. They show that over Switzerland the frequency of very hot days exceeding the 99th percentile of daily maximum temperature has more than tripled. Also, TXx in north-western Europe has a strong trend, as shown by Min et al. (2013). However, these studies also show that in many regions, such as eastern North America and western Europe, there are large discrepancies between modelled and observed trends in heat waves.

We propose three plausible mechanisms for the lack of a significant trend in TXx in India. The first is decadal variability. The second is a masking due to a trend in aerosols, i.e., worsening air pollution that causes less sunshine to reach the ground and thus a cooling influence, especially in dry seasons. This happened in Europe up to the mid-1980 (e.g., van Oldenborgh et al., 2009) and there is evidence that this plays a role in India (Krishnan and Ramanathan, 2002; van Donkelaar et al., 2015; Padma Kumari et al., 2007; Wild et al., 2007). The third mechanism is an increase in irrigation (Ambika et al., 2016) that leads to higher moisture availability and hence increased evaporation, leaving less energy to heat the air. This has been shown to decrease temperatures in California (Lobell and Bonfils, 2008) and India (Lobell et al., 2008; Douglas et al., 2009; Puma and Cook, 2010). We investigate each of these plausible mechanisms qualitatively in the next sections.



## 4   Decadal variability

The Indian Ocean has very little natural variability, with the trend dominating (Fig. 3). El Niño clearly plays a role, with the 5-month lagged Niño3.4 index explaining about a quarter of the remaining variance after subtracting the trend (as a regression on the smoothed global mean temperature). However, there is no decadal variability visible, especially after the second world

war when the quality of observations is higher. Considering well-known modes of decadal variability, the Pacific Decadal Oscillation (PDO) seems to cause higher temperatures along the Indian coasts, but this is just the effect of El Niño that is also visible in the PDO. The Atlantic Multidecadal Oscillation (AMO) does not have teleconnections to India. We conclude that decadal variability is very unlikely a cause of the lack of trend in TXx over India since the 1970s.

## 5   Aerosols

It is known that aerosols contribute to solar dimming, e.g., the reduction of solar radiation reaching the earth's surface (Streets et al., 2006; Wild et al., 2007). Krishnan and Ramanathan (2002) showed that the dry season trend is lower than for the wet season and ascribed the difference to the strongly increasing aerosol emissions. Padma Kumari et al. (2007) quantified the average solar dimming in India, and showed that the maximum temperatures over India are increasing at a much lower speed than expected from global warming, while minimum temperatures did increase at higher speed. For Jodhpur the reduction in

solar radiation reaching the surface between 1981 and 2004 was about $-1\,\mathrm{W m^{-2} yr^{-1}}$ in the predominantly cloud-free pre-monsoon months, while for Visakhapatnam in Andhra Pradesh the reduction was even more pronounced over these decades with about $-1.9\,\mathrm{W m^{-2} yr^{-1}}$.

The people in South-Asia and most of all the inhabitants of the cities in northern India suffer all year round from very high levels of air pollution. Expressed in terms of particle pollution (PM10) the annual mean may exceed the WHO 24 hr warning

levels for unhealthy conditions of 150 microgram per cubic meter (WHO, 2016). At ground level the pollution peaks in the winter season under an inversion layer, with a secondary peak in the pre-monsoon season, just before the aerosols are washed out at the onset of the monsoon precipitation. The effects on temperature are described by the aerosol optical depth (AOD), which includes the dimming effect of aerosols throughout the atmospheric column. The larger the AOD, the lower the fraction of sunlight that reaches the ground. In contrast to the ground-level concentrations, the AOD peaks at the monsoon onset in June

and is minimal in winter, when the air pollution is confined to a thin layer near the ground. Recently, Govardhan et al. (2016) reported on the observed and modelled differences between the high pre-monsoon (May) aerosol optical depth (AOD) and much lower post-monsoon (October) AOD over India, most notably over both study regions Rajasthan and Andhra Pradesh / Telangana (see their Figure 5). Aerosol components contributing to the high pre-monsoon AOD, though not well characterised, are thought to include significant amounts of black carbon, dust, and sea salt.

From ground-based observations it is reasonably well established that the AOD increased significantly before the 2000s, decreasing the incoming solar radiation and therefore giving rise to a cooling trend that opposes global warming (Padma Kumari et al., 2007). To study the changes in AOD spatially over India we use the MACC reanalysis (Bellouin et al., 2013), which is mainly constrained by MODIS AOD satellite observations. This reanalysis shows some improvements in some areas since



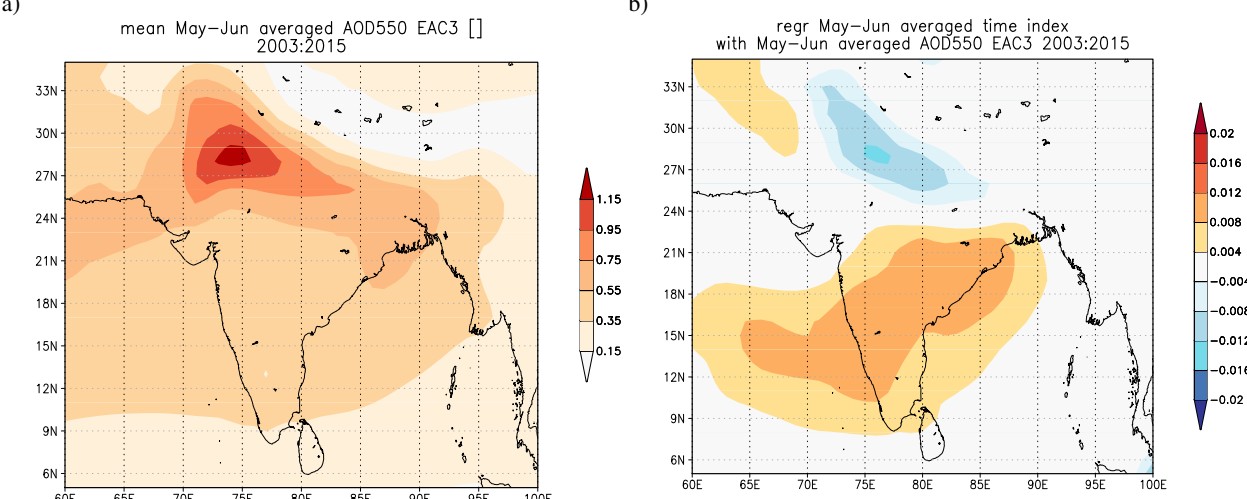

**Figure 7.** a) Mean May–June aerosol optical depth at 550 nm (AOD550) in the EAC3 reanalysis 2003–2015. b) Trend in AOD550 over this period ($yr^{-1}$).

2003, mainly in the northwest since the early 2000s (Fig. 7b). Spatially, there is some agreement between the area where AOD has started to recover over the MACC period and the area with more positive trends in TXx (compare Figs 4, 7b).

It is however still unclear to what extent the record maximum temperature in May 2016 could be related to a starting downward trend in AOD. The AOD at end of May 2016 still exceeded 1 over much of northern India (Fig. 7a), which is

the highest in the world outside deserts. In the region of Andhra Pradesh and Telangana the AOD still has a positive trend. A complicating factor for establishing trends in anthropogenic aerosol in the pre-monsoon period is the large interannual variability in dust load (Gautam et al., 2009). While dust storms might bring some relief by lowering maximum temperatures, these storms also exacerbate the health effects of heat waves. Also, high dust load related lower maximum temperatures during daytime would be accompanied during night by higher minimum temperatures through a reduction the outgoing longwave

radiation (Mallet et al., 2009), potentially compensating for the daytime dust-induced cooling. Because the observed total AOD during the heat waves of May 2016 in Rajasthan and May 2015 over Andhra Pradesh and Telangana were likely dust-dominated and not exceptionally low, the record maximum temperatures can not be attributed to an onset of solar brightening over these Indian regions.

To conclude, there is strong evidence that the increase in air pollution over India has given rise to a higher aerosol optical

depth in the pre-monsoon season, on top of year-to-year fluctuations in dust load which dominates the AOD in this season. The consequent reduction in surface solar radiation has resulted in a cooling trend in maximum temperatures during the pre-monsoon season, counteracting the warming trend due to greenhouse gases. There is no evidence that this trend has already reversed in the pre-monsoon period (Babu et al., 2013).



## 6 Moisture

Soil moisture plays an important role in altering the partitioning of the energy available at the land surface into sensible and latent heat fluxes. If the soil is dry, all incoming energy is used for heating the air temperature. Therefore, irrigation can play an important role in heat waves, making the soil wetter and therefore increasing the latent heat flux and reducing the sensible heat flux. This leads to lower temperatures but higher humidity (e.g., Lobell et al., 2008; Puma and Cook, 2010). In a combined measure such as the wet bulb temperature, the effects counteract each other.

As a measure for humidity we investigate the climatology and trends in dew point temperature, which is a function of specific humidity. In large parts of India, including the region that is affected by the heat wave, there is an increase in dew point temperature in the pre-monsoon month of May in the ERA-interim reanalysis (Fig. 8a). This increase could be due to expanded irrigation, although higher SST seems to play a role on the Pakistan coast. This agrees with Wehner et al. (2016), who find a significant increase in their heat index that also combines temperature and humidity, both in Karachi, Pakistan and Hyderabad, India.

Also, during an extremely hot period, humidity is very important for human health. In this sense, irrigation can have a negative effect on human health. The trend in humidity we found above is accompanied by a trend in wet bulb temperature in May, see also Fig. 8b). A positive trend in wet bulb temperature means that for the same high temperature in the past, the impact on people can be larger. The lack of trend in the highest temperature of the year, TXx, therefore does not imply that there is no increase in the severity of the impacts of heat waves.

The humidity of the pre-monsoon season has increased in large parts of India. Wehner et al. (2016) show that humidity increases due to rising SSTs. Another factor is the increase in irrigation in India over the last decades, the inland regions with largest humidity increases in Fig. 8a coincide with areas with increased irrigation (Lobell et al., 2008), who claim that the increase in irrigation already causes enough cooling to counteract greenhouse warming in northwestern India. In all regions with increased irrigation the resulting increase in evaporation counteracts the temperature trend due to global warming, but the increased humidity also makes some impacts of heat waves more severe.

## 7 Global coupled models

We next turn from observations and reanalyses to climate models. First we analyse TXx in the CMIP5 ensemble (Taylor et al., 2011) using the data from Sillmann et al. (2013) at the grid point closest to 26°N, 73°E corresponding to the area of the 2016 heat wave in Rajasthan. The MIROC models were excluded as these have unrealistically high temperatures in arid regions, reaching almost 70 °C here with large variability. A histogram of the trends in TXx in the other 22 models over 1975–2015 is shown in Fig. 9. When a model has $N_{ens}$ ensemble members these are each given weight $1/N_{ens}$ so that each model is weighted equally. Natural variability (estimated from intra-model variability) and model spread contribute about equally to the spread of the results. The mean trend is lower than other semi-arid areas at similar latitudes.

We compare this with the trends in the observed series at Bikaner (corrected for the spurious trend due to the decreasing amount of missing data) and Jodhpur, and the nearest grid point in ERA-interim. For reference we also show the trend in



a)                                                                    b)

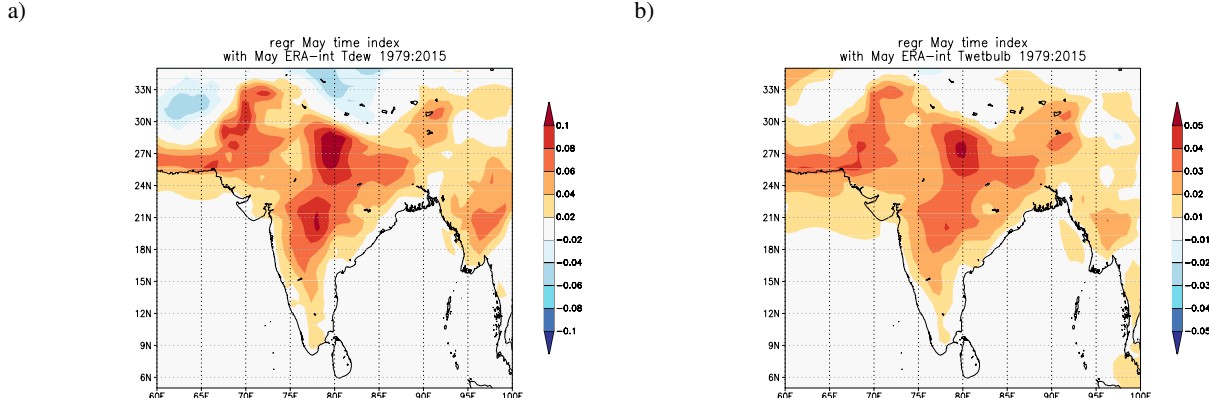

**Figure 8.** Trend in a) dew point temperature and b) wet bulb temperature in ERA-interim over May (K/yr).

the IMD Tx analysis, which is much higher but has not been corrected for the varying fraction of missing data and hence interpolation.

The only two ensemble members with a negative trend are from the CSIRO Mk3.6.0 and CCSM4. Other ensemble members of these models have much higher trends, so we ascribe the low values to natural variability. None of the models reproduces
the negative trends in the observed series and ERA-interim. The spatial pattern of the trend in TXx (Fig. 9b) also shows higher trends than observed over most of South Asia and the western Bay of Bengal (Fig. 4).

As the modelled trends are not compatible with the observed trends we did not use this ensemble for further analysis. (Similar problems were reported for the months of November–December in van Oldenborgh et al., 2016). The uncertainties in the representation of the effects of aerosols in the CMIP5 models are of course well-known (see, e.g., Bindoff et al., 2013,
and references therein) and trends in irrigation have only started to be included after CMIP5 (Wada et al., 2016). The large influence that these two misrepresented anthropogenic factors have on trends in TXx in India may well explain the discrepancy.

## 8   Atmosphere-only model

We furthermore used the distributed computing framework climate*prediction*.net to produce a very large ensemble of atmosphere-only general circulation simulations of the May 2016 in two different ensembles. The first ensemble (Actual) simulates possible
weather in the world we live in using current green-house gases and observed OSTIA SSTs and sea ice extent (Donlon et al., 2012) and a counterfactual ensemble (Natural) simulating possible weather in the world that might have been without anthropogenic climate change. As per Schaller et al. (2014), the anthropogenic signal was removed from the SSTs used to force the counterfactual simulations. As the uncertainty in estimating the pattern of anthropogenic warming is large, 11 different CMIP5 models and their multi-model mean were used to produce 12 different patterns of anthropogenic change in SSTs (see Table
S20.2 and Fig. S20.2 in Schaller et al., 2014), which we subtracted from the 2016 observations. We also looked at a third ensemble, climatological simulations of the years 1985-2014 using observed SSTs and sea ice extent from the OSTIA data



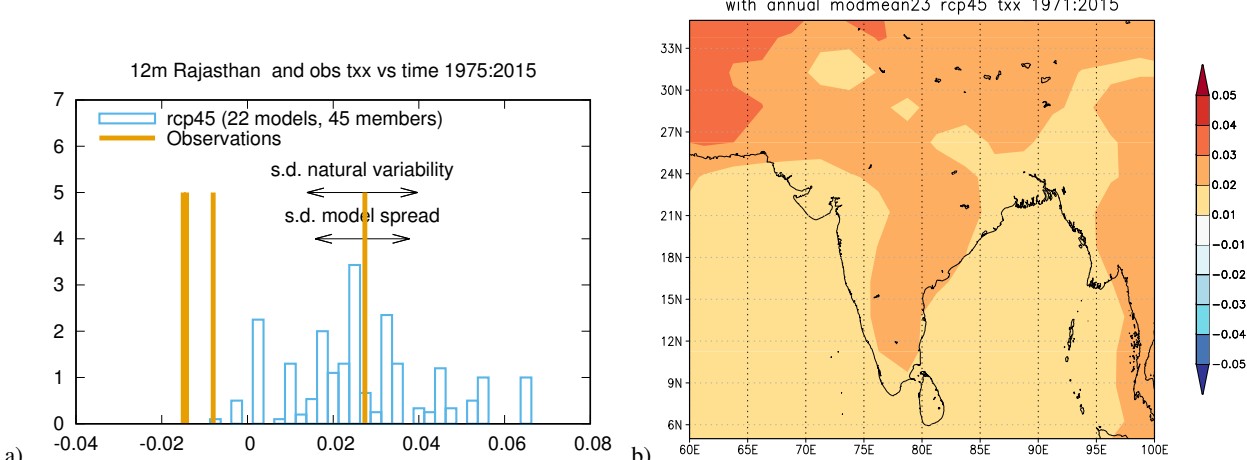

**Figure 9.** a) Histogram of the trend in TXx (K/yr) at the grid point closest to 26°N, 73°E in the CMIP5 ensemble (excluding the MIROC models). Ensemble members of the same model are downweighted so that each model gets weight one. This is compared with observations: the trends in Bikaner (corrected for varying missing data), Jodhpur and the nearest grid point of the ERA-interim reanalysis (left three bars). For reference the trend in the IMD gridded dataset without correction for varying missing data is also given (right-most bar). b) Mean of trend in TXx (K/yr) in the CMIP5 ensemble 1971–2015, with historical experiments up to 2005 extended with the RCP4.5 simulations 2006–2016.

set. These experiments are part of the weather@home project (Massey et al., 2015), and use the Met Office Hadley Centre atmosphere-only regional model HadRM3P at 50 km resolution over the CORDEX South Asia region, nested within the global circulation model HadAM3P at 1.875°×1.25° resolution.

Comparing the trend of TXx in the weather@home simulations in Fig. 10a to the observed and reanalysis trends in Fig. 4,
we find that the patterns agree to some extent in northwestern India where the 2016 heat wave occurred, but disagree sharply along the eastern coast where the 2015 heat wave took place. We suspect that the two factors suppressing the trend in TXx, aerosols and irrigation, are not well represented in this climate model (for example HadRM3p includes sulphate aerosols and the sulphur cycle but not black carbon). Nevertheless, we tentatively use this model for further analysis of the 2016 heat wave.

Fig. 10b shows the return periods of TXx in the region of the 2016 record in all three ensembles. The climatological
ensemble (orange) has a return period of a 51°C event occurring in May to be between 40 to 49 years (5–95% uncertainty range). This estimate compares well with return periods obtained from the observational analysis. Comparing this analysis with the simulations of the year 2016 under current climate conditions (red curve) we find that the return period for an event of the same magnitude is only 1 in 7 to 10 years, while the return period in the 'world that might have been' (blue curve) is 1 in 20 to 30 years.

In other words, the anthropogenic signal, as far as represented in the model approximately trebled the likelihood of the heat wave to occur, but the large scale teleconnection patterns as represented by the observed SSTs increased the likelihood of the event occurring by at least factor of four. This suggests that the particular large scale conditions of 2016 are a major driver of





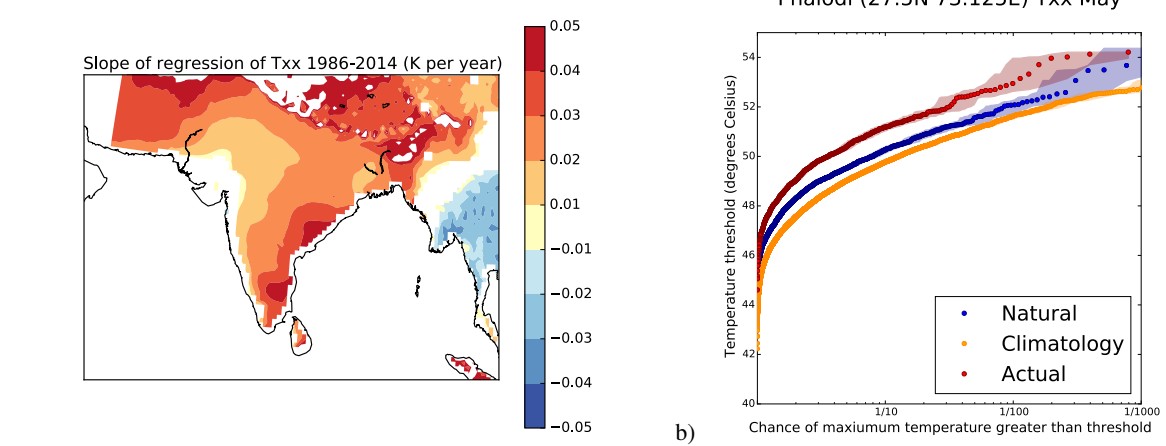

**Figure 10.** a) HadRM3P model trend 1986–2014 of TXx in May (K/yr). Ocean points and areas not significant at $p<0.05$ are shown as white. b) Return time plots for TXx in May at the grid box closest to Phalodi, for Actual 2016 simulations (red), Natural 2016 simulations (blue) and Climatology 1986-2014 simulations (yellow).

the record to be broken. In view of the still-large differences between the observed and modelled trends, probably stemming from the $\Delta SST$ from coupled models, misrepresented aerosol effects and lacking trends in irrigation, the factor of about three increase due to anthropogenic emissions has a large model uncertainty. We can therefore not make an attribution statement based on these model results.

## 9   Discussion

Contrary to most other regions of the world we find only limited evidence for positive trends in the highest temperature of the year (TXx) in India in observations and the ERA-interim reanalysis. The observed trends have a spurious component due to a decreasing fraction of missing data: more heat waves were missed in the 1960s and 1970s than the last decades, giving the appearance of a positive trend. This lack of trend implies that the recent record heat on 19 May 2016 in Rajasthan of 51 °C cannot simply be attributed to global warming. It was a fairly rare event with a local return period of more than 40 years. The heat wave in Andhra Pradesh and Telangana around 22 May 2015 that had a large human toll with thousands reported dead similarly cannot be connected to the global warming trend. This event was much less rare, with a return period of only about 15 years locally. One expects a heat wave with such a return period almost every year somewhere in India.

We investigated qualitatively three factors that could have counteracted the warming trend due to increased greenhouse gases. The first is decadal variability, for which we could find no evidence in the observed record. Well-known modes of decadal variability also do not have an expression in pre-monsoon heat in India.

The second factor is air pollution: the thickening 'brown cloud' over India prevents an increasing fraction of the sunlight from reaching the ground, leading to a negative trend in maximum temperatures. There is indeed evidence from both ground-





based and satellite observations that the aerosol optical depth has increased over the last decades. This increase may have reversed lately in a small region of northern India that is similar to the region with positive TXx trends. However, there is no evidence that the recent heat waves were already made more likely by improved air quality. Large interannual variability in the dust load during the pre-monsoon period will mask any long-term trend in anthropogenic pollution levels for some time.

We expect that the health and economic costs of the air pollution will make it necessary to regulate air quality. Air pollution exacerbates the health effect of heat waves. Besides the obvious benefits, a reduction in air pollution will lead to even higher maximum temperatures during heat waves.

    The third factor is increased humidity, which can result from more water availability, higher evaporative cooling and hence lower maximum temperatures. There is evidence for increased humidity during the pre-monsoon season in much of India. Part
of this is caused by higher sea surface temperatures, but the inland increases seem connected to increased use of irrigation in that season. Lobell et al. (2008) argue that the increase in northwestern India has been strong enough to counteract the warming trend from greenhouse gases. Although increased evaporation suppresses maximum air temperatures, an increase in humidity does increase health risks, notably in making the cooling of the human body by evaporation of sweat from the skin more difficult and hence increasing the risk over overheating. Unlike the aerosol cooling, evaporative cooling due to irrigation
is expected to increase, albeit at a reduced rate. Both this increase and higher SST will give an increase in humidity, leading to higher heat wave impacts.

    Next we considered how trends in heat waves in India are represented in climate models. The global coupled climate models used for the IPCC Fifth Assessment Report (Stocker et al., 2013) have trouble representing the lack of trends in the highest temperature of the year. The cooling factors mentioned above do not seem to be well-represented (aerosols) or are not included
(irrigation) in these models, which can therefore not be used for attributing the heat waves. An SST-forced model has a better representation of trends in northwestern India, but not along the east coast. Taken at face value, this model shows a factor three increase in probability for the 2016 heat wave due to anthropogenic factors, but with a large uncertainty again due to uncertainties in the representation of aerosols and lack of irrigation. This model also shows that the SST patterns of 2016 made the heat wave about a factor four times more likely in 2016 than in other years, showing potential seasonal predictability of the
event.

    This analysis also allows us to make qualitative projections for the near future, up to 2050. Global warming is set to continue until then with only minor differences between emission scenarios (Kirtman et al., 2013). $SO_2$ emissions averaged over India as a proxy for air quality are assumed to peak around 2020 in RCP2.6, 2030 in RCP4.5 and 2040 in RCP8.5 (Nakicenovic et al., 2000) and decline afterwards. With an expected doubling of India's energy demand up to 2040 (www.worldenergyoutlook.org/
india), only with prolonged and increasingly stringent air quality policies present-day air pollution levels would be mitigated in the coming decades (Cofala et al., 2015), so the peak date is still highly uncertain. Irrigation is projected to increase, though at a lower rate (Lobell et al., 2008) due to groundwater depletion (Wada et al., 2010). Humidity increases due to higher SST will also continue.

    For maximum temperatures, this means that the main heating factor, greenhouse warming, will continue. The cooling effect
of aerosols up to now will turn into an extra heating factor as the concentrations drop and the cooling effect of irrigation




is expected to diminish in the future. The result is projected to be a sharp increase in maximum temperatures over the next decades. (A similar effect was seen in Europe in the 1980s when air pollution was reduced, van Oldenborgh et al., 2009). Even though there has been no trend in TXx up to now, we expect a strong trend in the future.

Even without a discernible trend in temperature, the impacts of heat waves are considerable already. They have also risen due to the increasing air pollution and humidity during these heat waves, which both exacerbate the impacts of extreme heat on human health (Katsouyanni et al., 2009). The humanitarian impact of these heat waves was large, with 40% of all extreme weather related deaths attributed to heat waves in 2016, the largest proportion of total deaths of any type of extreme weather event (India Meteorological Department, Climate Research and Services, 13 January 2017). Experience in implementing heat wave interventions has shown that these deaths can be greatly reduced by straightforward measures such as keeping parks and homeless shelters open on the hottest days and providing early warning of a forecasted heat wave (Fouillet et al., 2008; Tran et al., 2013).

Changes in the vulnerability of the Indian population will ultimately determine the impact of future heat waves. As Indian cities continue their rapid growth, the number of people exposed to extreme heat due to inadequate housing and susceptible to heat-related illness due to lack of access to drinking water or electricity is set to increase (Taru Leading Edge, 2016; Tran et al., 2013). They will also face hotter and more humid heat waves.

## 10   Conclusions

There has been an all time Indian temperature record in Phalodi, Rajasthan, India on 19 May 2016. We investigated the influence of anthropogenic factors on the 2016 heat wave in Rajasthan and the 2015 heat wave in Andhra Pradesh and Telangana, which although no record had a large humanitarian impact. The 2016 event was rare, at least one in 40 years, but the 2015 event fairly common, about 1 in 15 yr.

Considering a meteorological definition of a heat wave, the highest temperature of the year TXx, we find no significant trend for either event. In agreement with earlier studies we find positive trends in this measure only in a limited area in northern India, and these are overestimated due to trends in missing data. The lack of trends is not due to decadal variability, but the increase in aerosols from air pollution and an increase in evaporation due to irrigation countered the upward trend due to greenhouse gases. Both these factors also increase the health risks of heat waves, so that the impacts have increased even if the temperatures have not.

The coupled climate models used in the latest IPCC report do not represent the trends in the highest temperature of the year correctly, and hence cannot be used to attribute the trend or project the future. An SST-forced model does little better. (For the 2016 heat wave it does find that the large scale climate conditions represented in the observed SSTs alone increased the likelihood by more than a factor of four.) We therefore do not consider it possible to give an attribution statement based on climate models for heat waves in India as measured by maximum temperatures.



However, this may not be the most relevant definition. Even without a discernible trend in temperature, the impacts of heat waves are considerable and have risen due to the combination with increasing air pollution and humidity. Experience has shown that exposure and vulnerability can be reduced substantially with straightforward measures.

For the near term future (up to 2050) we expect the trend due to global warming to continue, but the cooling effect of aerosols to diminish as air quality controls are implemented, diminishing its impact on health. Humidity will probably continue to rise as irrigation expands further, albeit at a reduced rate, and SST rises. The combination will give a strong rise of the temperature of heat waves. The high humidity will make health effects worse, whereas decreased pollution would decrease the impacts.

*Acknowledgements.* This research was done as part of the Raising Risk Awareness project — a partnership between the World Weather Attribution (WWA) Initiative and the Climate & Development Knowledge Network (CDKN) and also supported in part by the EU project EUCLEIA under Grant Agreement 607085. We would like to thank all of the volunteers who have donated their computing time to weather@home, and our colleagues at the Oxford eResearch Centre and the Met Office Hadley Centre PRECIS team for their technical expertise and scientific support.



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
