# Peer review of "Extreme heat in India and anthropogenic climate change"

_Natural Hazards and Earth System Sciences, 2017_

## Referee Comment (RC1) · F. Ambatlle (Referee) · 1 May 2017

The subject of the article is very interesting and provides a sound climate data base. However, there are 2 issues where I found room for improvement: To evaluate the impact on mortality due to extreme heat: absolutely numbers or adjectives ( " very lethal") are not the clearest ways to express it. The death figures is likely that they are much higher than the registered as heat related illness is often recorded inaccurately and figures from rural areas are hard to attain. To provide a sound evidence of the real impact I recommend to use variations on the mortality rate occurred during the two extreme heat events analysed. This rate could support sentences as "  "The previous year, a widely-reported very lethal heat wave occurred in the southeast, in Andhra Pradesh and Telangana, killing thousands of people "(page 1, l 2-3)  "From a health impact point of view, the severity of heat waves has increased in India" (page

1, l 13)

Although the article insists in the impact on mortality during heat waves, I would recommend analysing the increase of mortality during extreme hot single days. A study conducted in Catalonia concluded 40

The relation of air pollution and temperature is partly addressed: although some air pollutants have a cooling effect, others are climate forcers, which have a potential impact on the planet's climate and global warming in the short term (i.e. decades). Tropospheric O3 and black carbon (BC), a constituent of PM, are examples of air pollutants that are short‑lived climate forcers and that contribute directly to global warming. (Air Quality in Europe 2015 EEA Report No 5/2015)

Therefore, revision is needed in sentences as:

 Decadal variability cannot explain this, but both increased air pollution with aerosols blocking sunlight and increased irrigation leading to evaporative cooling have counteracted the effect of greenhouse gases up to now. (page 1, l 6-9)

 For the next decades, we expect the trend due to global warming to continue, but the cooling effect of aerosols to diminish as air quality controls are implemented. The expansion of irrigation will likely continue, though at a slower pace, mitigating this trend somewhat. Humidity will probably continue to rise. The combination will give a strong rise of the temperature of heat waves. The high humidity will make health effects worse, whereas decreased air pollution would decrease the impacts (page 1, l 14-17)

 The second is a masking due to a trend in aerosols, i.e., worsening air pollution that causes less sunshine to reach the ground and thus a cooling influence, especially in dry seasons. (page 10, l 25)

 Besides the obvious benefits, a reduction in air pollution will lead to even higher maximum temperatures during heat waves. (page 17, 5)

Homeless and outdoors professionals, should be added to the sentence Children and

the elderly are most vulnerable (page 2, l 5)

---

## Referee Comment (RC2) · A. Gershunov (Referee) · 10 Aug 2017

**SUMMARY**

This is a stimulating discussion paper by a very competent international/interdisciplinary team that is somewhat biased towards physical climate science while health science is lightly represented. Health impacts motivate this paper focused on analyzing Indian heat waves in various observational/reanalysis products as well as climate models including CMIP5 and targeted model experiments. It is found that the anthropogenic climate change signal is not so far detectable in the observed data on Indian heat waves. Evidence and reasoning are presented that suggest a global climate change trend masked by regional anthropogenic impacts, e.g. pollution

and irrigation. This suggestion is expressed in a somewhat speculative manner. The article is timely and the results are quite nuanced, but not as comprehensive as they possibly should be. In particular, the main definition of heat wave activity (hottest temperature of the year) is possibly not the most relevant definition for the tasks at hand and does not facilitate the fairest comparison between models and observations. Below I make specific comments and suggestions in the hope that they are useful in making the paper more comprehensive and making the results possibly more robust.

**SPECIFIC COMMENTS**

1. In addition to the daily maximum temperature (Tmax), the authors could examine the daily minimum temperature (Tmin). Tmin have been increasing more than Tmax over most regions around the globe. Pollution may be directly depressing Tmax trends via solar dimming in a region such as India while increased water vapor may be bolstering the Tmin trends. Tmin is also of very high relevance for heat wave impacts as inefficient nighttime cooling makes heat waves more difficult to weather and the elevated humidity responsible for it makes the Tmax expressions of heat waves more impactful on health. The latter signal may be partially accounted for by wet bulb temperature, though it is not clear how closely the two are correlated. Anyways, other regions have experienced the largest rise in heat wave activity via the Tmin expressions of heat waves, e.g. Gershunov et al. 2009.

Gershunov, A., D. Cayan and S. Iacobellis, 2009: The great 2006 heat wave over California and Nevada: Signal of an increasing trend. Journal of Climate, 22, 6181–6203.

2. The highest daily maximum (and minimum for that matter) temperature of the year is expected to vary a lot due to small sample size. If heat waves are becoming longerlasting or more frequent, that may not be well reflected in the annual maxima. A peaksover-threshold approach (Gershunov et al. 2009, Gershunov and Guirguis 2012) to quantify heat wave "activity", rather than its maximum day/year expression, would provide a larger sample size and would be more likely to show anthropogenic trends in both the Tmax and Tmin expressions of heat waves.

Gershunov A. and K. Guirguis, 2012: California heat waves in the present and future. Geophysical Research Letters, 39, L18710, doi:10.1029/2012GL052979.

3. Figure 3 shows the clearly increasing SST trend in the Northern Indian Ocean. This SST trend could be expected to relate to stronger heat wave activity, and particularly to more humid heat waves that would be expected to express more strongly in Tmin.

4. Reference to Alfaro et al. (2006) may be useful in the discussion on lines 24-31 on page 10. The relative contributions of SST and local soil moisture to summertime Tmin and Tmax are discussed quantified over a large part of North America.

Alfaro, E., A. Gershunov and D.R. Cayan, 2006: Prediction of summer maximum and minimum temperature over the Central and Western United States: The role of soil moisture and sea surface temperature. Journal of Climate. 19, 1407-1421.

5. The short section 4 that appears at the top of page 11 could benefit from a figure showing the average summertime Tmax (and Tmin). Since it is mentioned elsewhere in the text that these trends exist, it would be useful to see these time series at the stations used here.

6. The last paragraph on page 12 raises the question: Does the AOD trend reduce average summertime Tmax by the same factor as it reduces the extreme heat events' Tmax expression?

7. Section 6 at the top of page 13 could benefit from reference to Alfaro et al. (2006), who expressly quantify the impact of soil moisture on summertime Tmax over a large part of North America.

8. Section 6, last sentence: "...increased humidity also makes some impacts of heat waves more severe" and/as it tends to keep them hotter at night (Gershunov et al. 2009).

9. Discussion section, first paragraph. While it may be true that "the record heat on 19 May 2016 ... cannot simply be attributed to global warming", I wonder if the fact that "we find only limited evidence for positive trends in the highest temperature of the year in India in observations and the ERA-interim reanalysis" is enough to claim that there is no evidence of historical heat wave trends. As suggested above, a peaks-over-threshold approach may well lead to a clearer identification of historical trends (e.g. Gershunov et al. 2009).

10. The last sentence of the same paragraph states: "One expects a heat wave with such a return period almost every year somewhere in India". I wonder what is the basis for this statement? Does this argument hinge on the scale of heat waves relative tot hat of the Indian Subcontinent? A clarification would be useful here.

11. Page 17, lines 14-16. The increase in irrigation "and higher SST will give an increase in humidity, leading to higher heat wave impacts" also from elevated nighttime (Tmin) temperature (e.g. Gershunov et al. 2009).

12. At the bottom of page 17, the authors describe and try to explain the disconnect between models and observations particularly with respect to the hottest temperature of the year. On line 1 of page 19, the authors suggest this may not be the most relevant definition. I think this definition is making it difficult to detect a trend in observations, even if a trend in heat wave activity exists. It also makes it harder for the models to simulate the observed trend. Defining heat wave activity as temperature excesses over a high percentile (relative to local, observed and modeled climatology) threshold, would probably lead to a more relevant comparison that would also be fairer to the models. And it certainly makes it hard (without considering Tmin) to detect temperature trends related to changes in humidity, which, as the authors state, are so important for health impacts.

---

## Author Comment (AC1) · 12 Oct 2017

We thank the reviewer for her comments on our article on the attribution of heat waves in India. Below we answer her comments and questions and show how these have been taken into account in the revised text.

*The subject of the article is very interesting and provides a sound climate data base. However, there are 2 issues where I found room for improvement:*

*To evaluate the impact on mortality due to extreme heat: absolutely numbers or adjectives ( " very lethal") are not the clearest ways to express it. The death figures is likely that they are much higher than the registered as heat related illness is often recorded inaccurately and figures from rural areas are hard to attain. To provide a sound evi-*

[Figure]

*dence of the real impact I recommend to use variations on the mortality rate occurred during the two extreme heat events analysed. This rate could support sentences as 'The previous year, a widely-reported very lethal heat wave occurred in the southeast, in Andhra Pradesh and Telangana, killing thousands ofpeople' (page1,l2-3). 'From a health impact point of view, the severity of heat waves has increased in India'.*

The first sentence was only included to show that the impact of the heat waves was large. As the reviewer states, death figures from the rural areas were most deaths were (anecdotally) reported are not known, and tracing these and tying them to heat is a job for specialists. At this moment we cannot do more than state that there has been anecdotal impact of the heat waves on mortality, especially the 2015 Andhra Pradesh event.

The second sentence the reviewer quotes is based on generalities: although we do not find any evidence that the hottest day of the year has increased, the humidity and air pollution in these very hot days have increased so that the impact of these high temperatures must have increased. We updated that sentence to 'From these factors it follows that, from a health impact point of view, the severity of heat waves has increased in India.'

*Although the article insists in the impact on mortality during heat waves, I would recommend analysing the increase of mortality during extreme hot single days. A study conducted in Catalonia concluded concluded 40% of deaths due to heat, occurred in single days of extreme temperature. Xavier Basagaña et al. "Heat Waves and cause-specific mortality at all ages". Epidemiology. Volume 22, number 6, November 2011. DOI: 10.1097/EDE.0b013e31823031c5.*

Indeed we use the single-day maximum temperature and maximum wet bulb temperature for this study, based on the anecdotal observation that most casualties were outdoor labourers that succumbed to heat stress. We thank the reviewer for the reference, of which we were not aware, and have added it to the text:

'The one-day length of the definition was chosen because of anecdotal evidence that the main victims in the 2015 heat wave were outdoor labourers. Basagaña et al. (2011) also find no stronger effect from longer heat waves in Catalonia. In contrast, in urban areas it is often found that longer heat waves have larger impacts, as the heat takes some time to penetrate the buildings to the most vulnerable population (e.g., Tan et al., 2007; D'Ippoliti et al., 2010).'

*The relation of air pollution and temperature is partly addressed: although some air pollutants have a cooling effect, others are climate forcers, which have a potential impact on the planet's climate and global warming in the short term (i.e. decades). Tropospheric O3 and black carbon (BC), a constituent of PM, are examples of air pollutants that are shortâA?S?lived climate forcers and that contribute directly to global warming. (Air Quality in Europe 2015 EEA Report No 5/2015).*

While it is true that pollutants have both cooling and warming effects, the incident solar radiation at the surface is reduced by all scatterers and absorbers in the atmosphere. The net climate effect of India's air pollution is estimated to have cooled the Indian subcontinent with up to -0.3 degrees since 1970s (Krishnan and Ramanathan, 2002) and thus reduced daily maximum 2-m temperatures. At the same time atmospheric heating is increased. Krishnan and Ramanathan suggest that the atmospheric warming is being distributed regionally and could contribute to reduced tropical precipitation. While such wider climate effects of the atmospheric heating are very uncertain, the surface cooling below the haze layers relevant in this study on heatwaves is undisputed. To better explain that we are referring to a surface cooling effect (instead of climate cooling) by both scattering and absorbing aerosols 'cooling' is replaced by 'surface cooling' at a couple of places in the paper. Furthermore we added a few sentences in the aerosol section: 'Note that all types of aerosols block part of the incident sunlight and thus cool the surface, decreasing maximum temperatures. Absorbing aerosols additionally heat the lower atmosphere and are thought to affect the regional climate through changes in cloudiness and tropical precipitation (Krishnan and Ramanathan,

2002). The redistribution of the enhanced atmospheric heating by black carbon is still poorly understood.'.

*Therefore, revision is needed in sentences as:*

- *Decadal variability cannot explain this, but both increased air pollution with aerosols blocking sunlight and increased irrigation leading to evaporative cooling have counteracted the effect of greenhouse gases up to now. (page 1, l 6-9)*

  Unchanged.

- *For the next decades, we expect the trend due to global warming to continue, but the cooling effect of aerosols to diminish as air quality controls are implemented. The expansion of irrigation will likely continue, though at a slower pace, mitigating this trend somewhat. Humidity will probably continue to rise. The combination will give a strong rise of the temperature of heat waves. The high humidity will make health effects worse, whereas decreased air pollution would decrease the impacts (page 1, l 14-17)*

  Text change: 'the cooling effect' has been changed to 'the surface cooling effect'

- *The second is a masking due to a trend in aerosols, i.e., worsening air pollution that causes less sunshine to reach the ground and thus a cooling influence, especially in dry seasons. (page 10, l 25)*

  Text change: 'a cooling influence' has been changed to 'a surface cooling influence'

- *Besides the obvious benefits, a reduction in air pollution will lead to even higher maximum temperatures during heat waves. (page 17, 5)*

  Text not changed

- *Homeless and outdoors professionals, should be added to the sentence Children and the elderly are most vulnerable (page 2, l 5)*

Indeed, thank you for the suggestion.

**References**

Basagaña, X., Sartini, C., Barrera-Gómez, J., Dadvand, P., Cunillera, J., Ostro, B., Sunyer, J., and Medina-Ramón, M.: Heat Waves and Cause-specific Mortality at all Ages, Epidemiology, 22, 765–772, doi:10.1097/EDE.0b013e31823031c5, 2011.

D'Ippoliti, D., Michelozzi, P., Marino, C., de'Donato, F., Menne, B., Katsouyanni, K., Kirchmayer, U., Analitis, A., Medina-Ramón, M., Paldy, A., Atkinson, R., Kovats, S., Bisanti, L., Schneider, A., Lefranc, A., Iñiguez, C., and Perucci, C. A.: The impact of heat waves on mortality in 9 European cities: results from the EuroHEAT project, Environmental Health, 9, 37, doi: 10.1186/1476-069X-9-37, 2010.

Krishnan, R. and Ramanathan, V.: Evidence of surface cooling from absorbing aerosols, Geophys. Res. Lett., 29, 54–1–54–4, doi:10.1029/2002GL014687, 2002.

Tan, J., Zheng, Y., Song, G., Kalkstein, Laurence, S., Kalkstein, A. J., and Tang, X.: Heat wave impacts on mortality in Shanghai, 1998 and 2003, Int. J. Biometeorol, 51, 193–200, doi:10.1007/s00484-006-0058-3, 2007.

---

## Author Comment (AC2) · 12 Oct 2017

We thank the reviewer for his useful comments and suggestions that helped to improve the paper.

*This is a stimulating discussion paper by a very competent international/ interdisciplinary team that is somewhat biased towards physical climate science while health science is lightly represented. Health impacts motivate this paper focused on analyzing Indian heat waves in various observational/reanalysis products as well as climate models including CMIP5 and targeted model experiments. It is found that the anthropogenic climate change signal is not so far detectable in the observed data on Indian heat waves. Evidence and reasoning are presented that suggest a global climate*

*change trend masked by regional anthropogenic impacts, e.g. pollution and irrigation. This suggestion is expressed in a somewhat speculative manner. The article is timely and the results are quite nuanced, but not as comprehensive as they possibly should be. In particular, the main definition of heat wave activity (hottest temperature of the year) is possibly not the most relevant definition for the tasks at hand and does not facilitate the fairest comparison between models and observations. Below I make specific comments and suggestions in the hope that they are useful in making the paper more comprehensive and making the results possibly more robust.*

Although many elements of heat waves in India have been discussed before, we think this is the first time they have been put together. We agree that the reasoning is qualitative at the moment, but did not have the resources to do a full quantitative study at this moment.

1. *In addition to the daily maximum temperature (Tmax), the authors could examine the daily minimum temperature (Tmin). Tmin have been increasing more than Tmax over most regions around the globe. Pollution may be directly depressing Tmax trends via solar dimming in a region such as India while increased water vapor may be bolstering the Tmin trends. Tmin is also of very high relevance for heat wave impacts as inefficient nighttime cooling makes heat waves more difficult to weather and the elevated humidity responsible for it makes the Tmax expressions of heat waves more impactful on health. The latter signal may be partially accounted for by wet bulb temperature, though it is not clear how closely the two are correlated. Anyways, other regions have experienced the largest rise in heat wave activity via the Tmin expressions of heat waves, e.g. Gershunov, A., D. Cayan and S. Iacobellis, 2009: The great 2006 heat wave over California and Nevada: Signal of an increasing trend. Journal of Climate, 22, 6181– 6203.*

   Following this suggestion we have added analyses of Tmin trends to the paper. There is a problem that the annual maximum of Tmin tends to be later in the

seasonal cycle, around the onset of the monsoon, and hence heat waves defined in Tmax usually do not correspond to annual maxima of Tmin. We minimise, but do not eliminate this, by considering only May TNx. Even then, the heat waves considered in this article do not stand out in Tmin, so we only discuss Tmin in the text in the context of the specific heat waves (Fig. 1).

The station data analysis could not be done for Tmin in the 2016 heat wave, as none of the stations considered (Bikaner, Jodhpur and Machilipatnam) has enough valid Tmin data to do a sensible analysis. The amount of missing data is much higher in the Tmin than Tmax. A sentence to this effect has been added to the text.

We added trend map of TNx and Tmin in may (new Fig. 9, also attached to this response). These show that there are clear positive trends in the areas of strongest increases in irrigation in the Ganges valley, supporting our argument that this is a factor there in suppressing Tmax trends. This has been added to the text:

'The trends in the highest May minimum temperature (TNx) show strong increases in TNx around New Delhi and in the Punjab (Fig. 9a,b) and positive trends in the whole Ganges valley. The May averaged TN gives basically the same pattern, Fig. 9c,d. This gives additional support for the role of irrigation, as these are the areas where irrigation has increased. The increased evaporation and humidity are expected to increase night-time temperatures as water vapour is a very effective greenhouse gas (e.g., Gershunov et al., 2009). The increase humidity and wet bulb temperature in central India are not reflected in increased minimum temperatures there. We do not know the reasons, and note again that the redistribution of the enhanced atmospheric heating by black carbon is still poorly understood.'

2. *The highest daily maximum (and minimum for that matter) temperature of the year is expected to vary a lot due to small sample size. If heat waves are becoming*

*longer-lasting or more frequent, that may not be well reflected in the annual maxima. A peaks- over-threshold approach (Gershunov et al. 2009, Gershunov and Guirguis 2012) to quantify heat wave "activity", rather than its maximum day/year expression, would provide a larger sample size and would be more likely to show anthropogenic trends in both the Tmax and Tmin expressions of heat waves. Gershunov A. and K. Guirguis, 2012: California heat waves in the present and future. Geophysical Research Letters, 39, L18710, doi:10.1029/2012GL052979.*

In our experience, POT and block maxima analyses have similar sensitivities for heat waves. These last a few days and only occur in a few weeks in May–June, so there are just not that many degrees of freedom per year. The annual maximum analysis has the advantage of being much simpler. In India there is very little decadal variability, in contrast to California, so that the number of degrees of freedom is equal to the number of years.

Another measure of heat wave, such as the activity proposed in Gershunov and Guirguis 2012, is more difficult to connect to the heat wave impact literature and also to the measures of heat waves that are being communicated at the moment, which is almost exclusively TXx.

Finally, w find that the main problem is India is not natural variability. We show that systematic errors due to aerosols and irrigation not being represented properly in models preclude us from using these to make quantitative projections. This conclusion would be the same whatever measure of heat wave we adopt.

3. *Figure 3 shows the clearly increasing SST trend in the Northern Indian Ocean. This SST trend could be expected to relate to stronger heat wave activity, and particularly to more humid heat waves that would be expected to express more strongly in Tmin.*

This is what we would expect from Gershunov et al. (2009). However, it is not what the IMD analysis and ERA-interim reanalysis show. there are no trends in

humidity nor in Tmin along the coasts (see Fig. 8 in the manuscript nd the new Fig. 9 that is attached. Apparently nature is not as simple as we would like to believe.

4. *Reference to Alfaro et al. (2006) may be useful in the discussion on lines 24-31 on page 10. The relative contributions of SST and local soil moisture to summertime Tmin and Tmax are discussed quantified over a large part of North America. Alfaro, E., A. Gershunov and D.R. Cayan, 2006: Prediction of summer maximum and minimum temperature over the Central and Western United States: The role of soil moisture and sea surface temperature. Journal of Climate. 19, 1407-1421.*

We are discussing trends in this paragraph: not seasonal predictability based on variations of SST (ENSO) or soil moisture due to rainfall, but trends in heat waves due to trends in SST and irrigation. (It should be noted that there are also hardly any ENSO teleconnections to pre-monsoon Tmax in India.)

5. *The short section 4 that appears at the top of page 11 could benefit from a figure showing the average summertime Tmax (and Tmin). Since it is mentioned elsewhere in the text that these trends exist, it would be useful to see these time series at the stations used here.*

We are not sure what is meant here. Time series of TXx at the stations are shown in Fig. 6. Average summertime Tmax or Tmin is ill-defined in India, as there are large differences between temperatures pre-monsoon (high Tmax), around the onset (high Tmin) and afterwards. We have added trend maps of pre-monsoon (May-June) Tmax and Tmin for reference to Figs 4 and 9, plus text referring to these panels. They show very similar patterns to TXx and TNx respectively and add additional evidence for the points we make.

6. *The last paragraph on page 12 raises the question: Does the AOD trend reduce average summertime Tmax by the same factor as it reduces the extreme heat events' Tmax expression?*

This is a good question. As the added plot shows the trends in TXx and Tmax averaged over May-June are similar, so the proposed effect of aerosols on these two time scales could also be comparable.

7. *Section 6 at the top of page 13 could benefit from reference to Alfaro et al. (2006), who expressly quantify the impact of soil moisture on summertime Tmax over a large part of North America.*

See above.

8. *Section 6, last sentence: "... increased humidity also makes some impacts of heat waves more severe" and/as it tends to keep them hotter at night (Gershunov et al. 2009).*

Indeed, thanks. Added.

9. *Discussion section, first paragraph. While it may be true that "the record heat on 19 May 2016 ... cannot simply be attributed to global warming", I wonder if the fact that "we find only limited evidence for positive trends in the highest temperature of the year in India in observations and the ERA-interim reanalysis" is enough to claim that there is no evidence of historical heat wave trends. As suggested above, a peaks- over-threshold approach may well lead to a clearer identification of historical trends (e.g. Gershunov et al. 2009).*

We argued before that this is not a case of natural variability drowning a signal, but of other factors counteracting the warming effect of greenhouse gases. We have the statistical power to detect a reasonable trend should one exist (taking care of the decrease in missing data). We added a sentence making this clearer: 'This is not due to natural variability on relatively short time series: the upper bounds on the trends are smaller than we would expect from other regions around the globe.' In our recent study of heat waves in the Mediterranean we typically found trends in TX3x (and TXx, unpublished) almost two times the global mean

trend, i.e., 0.3 K/10yr from 1979. This would be at the upper limit of the range at Bikaner, $3\sigma$ above the trend in Jodhpur and just below the upper bound at Machilipatnam. A separate line of evidence is that given the small decorrelation length of heat waves compared to the size of the Indian subcontinent, we would expect a noise pattern centered around 0.02 to 0.03 K/yr in the spatial trend. Instead we find a pattern centered around zero. This is confirmed in the new seasonal mean plots.

10. *The last sentence of the same paragraph states: "One expects a heat wave with such a return period almost every year somewhere in India". I wonder what is the basis for this statement? Does this argument hinge on the scale of heat waves relative to that of the Indian Subcontinent? A clarification would be useful here.*

Expanded to 'As the area covered by the heat wave was less than 1/15 of the area of India, one expects a heat wave with such a return period almost every year somewhere in India.' One has as many chances to get a heat wave as there are independent areas that have heat waves. The maps of Fig. 1 show that especially the 2015 heat wave was very localised, with an area much smaller than 1/15 of the area of India. The 2016 heat wave had about this size. The expected number of heat waves with the size and return time of the 2015 Andhra Pradesh event is thus more than one per year somewhere in India.

11. *Page 17, lines 14-16. The increase in irrigation "and higher SST will give an increase in humidity, leading to higher heat wave impacts" also from elevated nighttime (Tmin) temperature (e.g. Gershunov et al. 2009).*

These were detailed in the discussion, including the reference, we feel the level of abstraction here does not require this detail.

12. *At the bottom of page 17, the authors describe and try to explain the disconnect between models and observations particularly with respect to the hottest temperature of the year. On line 1 of page 19, the authors suggest this may not be the*

*most relevant definition. I think this definition is making it difficult to detect a trend in observations, even if a trend in heat wave activity exists. It also makes it harder for the models to simulate the observed trend. Defining heat wave activity as temperature excesses over a high percentile (relative to local, observed and modeled climatology) threshold, would probably lead to a more relevant comparison that would also be fairer to the models. And it certainly makes it hard (without considering Tmin) to detect temperature trends related to changes in humidity, which, as the authors state, are so important for health impacts.*

Regarding the first point, all evidence we have points to instantaneous high Tmax having the largest impact on the rural population, e.g., during the heat wave in Andhra Pradesh. This is also the number that is quoted most frequently in meteorological discussions of heat waves, for instance in the widespread attribution of the record temperatures in Phalodi to global warming. We therefore take TXx as a starting point. Excesses over a temporally varying climatology would give 'heat waves' in other seasons than the hottest pre-monsoon months that have very different impacts and are not reported by meteorologists. The impacts also do not increase strongly with the length of the heat wave as far as we know (unlike the European urban situation).

Secondly, we do not want to be fair to the models. We just demand that they reproduce trends in the quantity of interest, which they fail to do for fairly obvious reasons: missing trends in irrigation and underestimated aerosol cooling trends. These are well-known problems that should be remedied in CMIP6. However, we stress that models that do not take these factors into account cannot be used for attribution (or projections) of heat waves in India.

Finally, we do try to take into account humidity (though not aerosols) into our heat wave definition with a stronger link to impacts via the wet bulb temperature, which has been used in many publications. However, if the models cannot get the temperature trends nor the humidity trends right, should we trust these errors

to cancel to get the wet bulb temperature (or heat index) trends correct?

In the end we are forced to state qualitative projections based on the physical insights we have gained. We hope we or others can use this publication as a starting point to tune models to reproduce the observed trends, and so make quantitative projections.

**References**

Gershunov, A., Cayan, D. R., and Iacobellis, S. F.: The Great 2006 Heat Wave over California and Nevada: Signal of an Increasing Trend, Journal of Climate, 22, 6181–6203, doi:10.1175/ 2009JCLI2465.1, 2009.

[Figure]

**Fig. 1.**